# ICNIRP Guidelines’ Exposure Assessment Method for 5G Millimetre Wave Radiation May Trigger Adverse Effects

**DOI:** 10.3390/ijerph20075267

**Published:** 2023-03-27

**Authors:** Mary Redmayne, Donald R. Maisch

**Affiliations:** 1School of Geography, Environment and Earth Sciences, Victoria University of Wellington, Kelburn Parade, Wellington 6012, New Zealand; 2Oceania Radiofrequency Scientific Advisory Association Inc. (ORSAA), Brisbane, QLD 4020, Australia; 3The Australasian College of Nutritional and Environmental Medicine (ACNEM), Melbourne, VIC 3205, Australia

**Keywords:** electromagnetic radiation, safety hazards, exposure evaluation, public health, beamforming

## Abstract

The current global roll-out of 5G infrastructure is designed to utilise millimetre wave frequencies (30–300 GHz range) at data transmission rates in the order of gigabits per second (Gbps). This frequency band will be transmitted using beamforming, a new introduction in near-field exposures. The International Commission on Non-Ionising Radiation Protection (ICNIRP) has recently updated their guidelines. We briefly examine whether the new approach of the ICNIRP is satisfactory to prevent heat damage and other adverse bio-effects once millimetre wave 5G is included, and we challenge the use of surface-only exposure assessment for local exposures greater than 6 GHz in part due to possible Brillouin precursor pulse formation. However, this is relevant whether or not Brillouin precursors occur from absorption of either 5G or future G transmissions. Many significant sources conclude there is insufficient research to assure safety even from the heat perspective. To date, there has been no published in vivo, in vitro or epidemiological research using exposures to 5G New Radio beam-formed signals.

## 1. Introduction 

The current global roll-out of 5G infrastructure is designed to introduce and utilise millimetre wave frequencies (30–300 GHz range) at data transmission rates in the order of gigabits per second (Gbps). This frequency band will be transmitted using beamforming [1]. This paper examines the rigor and relevance of the International Commission for Non-Ionising Radiation Protection’s (ICNIRP) updated exposure guidelines regarding possible health impacts from exposure to 5G beam-formed emissions [2]. The ICNIRP guidelines seek to prevent heat and shock injury from radiofrequency radiation exposure over the very short term. 

The fact that millimetre wave (mmW) penetration is normally limited to the skin with minimal energy absorption imparted into the body has led to assumption that it is not necessary to measure or restrict this absorbed energy in the mmW bandwidth, but that surface restrictions are sufficient [2]. We explore in more detail the ways this assumption is flawed and suggest how it could be addressed.

5G (formally called 5G New Radio) uses a wide range of radio frequency bands as well as relying on fibre. Groupe Speciale Mobile Association (GSMA) has outlined the bands that providers should aim to use. These bands include low, mid, and high spectrum bands. The low-band spectrum will initially use existing bands of 900 MHz, 850/500 MHz and 700 MHz, with a view to adding the 600 MHz range, and ideally all low bands available. The mid band will include “as much contiguous spectrum as possible” in the 3.3–4.2 GHz band; also needed in mid band are 2.3 GHz and 2.5 GHz, and over time, more will be added between 3 and 24 GHz. The high band, also referred to as mmW bands, will include bands in the range of 26 GHz, 28 GHz and 40 GHz, with a view to incorporating 66–71 GHz “to encourage timely equipment support” [3]. 

The mmW band will be transmitted using beamforming—the first time this approach has been intentionally incorporated into telecommunication devices for public use, which will include near-field exposures from hand-held devices. Indeed, until 5G development, near-field exposures from this source were not investigated or considered in research or standards [4]. The 5G beam-forming component will be targeted to specific devices in narrow, high-power beams compared to 3G or 4G, which are transmitted routinely in all directions available to each antenna. Opinion is mixed about whether this will increase or decrease overall environmental exposures. However, during use, the energy in 5G beams will be relatively high for those in their path and those handling receiving/sending devices; the beamed energy will interact with people, trees and animals in its path. The latter includes pollinating insects for whom increased absorption between 3 and 370% can be expected if only 10% of incident power shifts to frequencies greater than 6 GHz [5]. 

This paper explores the assumption by ICNIRP that it is not necessary to assess 5G exposures > 6 GHz using specific absorption (SA_ab_).

## 2. Methods

We assessed new or changed aspects of the 2020 ICNIRP guidelines [2], particularly measurement approaches for exposures > 6 GHz and the ways that heat impact could occur. We searched the relevant scientific literature using the ORSAA and EMF-Portal databases and the search words ‘millimetre wave*’ (mmW) and/or ‘Brillouin’ combined with others including: heat*; beam-forming; 5G; absorb*; dermis. We describe and discuss Brillouin precursor pulses, and then present other relevant research.

## 3. Results

### 3.1. ICNIRP Guidelines

Coinciding with the development of 5G with its projected use of mmW, the ICNIRP guidelines were updated and published in 2020 [2]. They take a frequency-based approach to assessing exposures. For frequencies < 6 GHz, the basic restrictions use specific absorption rate (SAR), as previously, but add specific absorption (SA) to ensure that the cumulative energy absorbed over 6 min is not so rapid that it could result in “operational adverse health effects” (p. 493), meaning heat damage. Both consider mass, using W/kg and J/kg, respectively.

The updated ICNIRP guidelines remove the use of SAR and SA for exposures > 6 GHz since they claim, “above about 6 GHz this heating occurs predominantly within the skin” [5]. As a result, the basic restrictions for >6 GHz to 300 GHz exposure are to be assessed with absorbed power density (ICNIRP S_ab_), measured in W/m^2^, thus limiting assessment to measuring power absorbed by a surface. However, S_(ab)_ measurements are likely to be unreliable in the near field, as explained by Belyaev et al. [6]. 

For exposures > 6 GHz, to prevent overly rapid temperature rise, the ICNIRP specify measuring local absorbed energy density (U_ab_), also a surface measurement (J/m^2^). In both cases, any internal heating from absorbed energy is ignored. 

This is of concern because previous studies have demonstrated that mmW exposures can result in extremely high SAR readings even when the surface measures such as J/m^2^ are acceptably low [7]. The shallower the absorption of energy, the higher the risk of unacceptably high surface heating.

The updated guidelines have also introduced two categories of tissue, being those that are commonly cooler than core temperature and those that are not. Skin is included in the former category. The calculations and allowances for skin exposures assume that skin (and various other body parts) can safely be heated by 5 °C based on the normothermal [sic] skin temperature averaging < 33–36 °C; this is with the goal of keeping skin temperature rise below 41 °C, at which point tissue damage can occur, p. 489 [2]. Problems for those suffering fever or other temperature control health problems are claimed to be precluded by the safety margin imposed over the level at which ≥41 °C would occur [2] (p. 492) It needs to be borne in mind that skin modelling studies treat the skin as homogeneous dermal tissue although it is a complex organ interacting with the environment through nerve endings almost at the surface and blood vessels very near the surface. Further, the induced temperature increase can be underestimated by more than a factor of three [8].

Declared confidence in the safety of the 2020 ICNIRP approach led to Australia’s standards committee removing the precautionary clause from their new exposure standard (RPS-S1 (Rev.1)), saying it is unnecessary [9]. This is despite ICNIRP guidelines [2] and Dr Karipidis of the Australian Radiation Protection and Nuclear Safety Agency (ARPANSA) acknowledging that higher frequencies can transport heat deep within the body [9]. 

ARPANSA is the government body whose responsibility it is to provide advice and services on radiation protection to Australians. This aspect of their purview is led by Dr Karipidis, who is also a main commissioner with ICNIRP. 

### 3.2. Brillouin Precursors

Many countries have not yet introduced the mmW band into what is being called 5G by some and 4.5G by others. In these cases, beamforming is being transmitted at a mid- band around 3.5 GHz. Penetration of the focused energy at 6 GHz is around 8 mm [2] (p. 504). Since the penetration depth increases as the frequency drops, it will be deeper than this at 3.5 GHz. Skin is approximately 2 mm thick, so, under this scenario, all skin layers and near-surface blood vessels will be being heated by all 5G frequencies. 

The addition of using mmW frequencies for beamforming will reduce the depth of primary penetration, and structures nearest the skin’s surface will be preferentially heated in this frequency band. For instance, at 30 GHz, penetration is 0.92 mm (ICNIRP 2020 Table 10, p. 504) [2]. However, the upper layer of the dermis—the papillary dermis— and the nerve endings immediately in front of the ear are shallower (Table 1). At a depth of approximately 1.15–1.45 mm [10], the skin is rich in papillae, containing an extensive capillary network and neural structures (Table 1). Since penetration is approximately 0.23 mm at 300 GHz [2] (p. 504), even fully operational 5G will impact on upper cutaneous blood vessels and free nerve endings in some parts of the body (Table 1).

Most of the transmitted mmW energy is deposited in the tissues within a mm of the skin or eye surface. Using these and higher frequencies for 5G may create pulses that carry some of the energy more deeply into the body. 

Sommerfeld and Brillouin precursor pulses were eponymously predicted in 1914, and first demonstrated in 1969 [13]. 

The Sommerfeld pulse occurs immediately after the relevant light pulse’s arrival at a dispersive dielectric surface, such as skin. It displays a signal that is a considerably higher frequency than that originally generated, but it is weak and declines rapidly. It is not thought to be of concern in the current scenario.

However, directly after the Sommerfeld pulse, another precursor (also known as forerunner) pulse appears, this time starting well below the transmitted frequency but rising very rapidly; if the incident wave has a sudden start, this second precursor covers the full spectrum of frequencies from −∞ to ∞ [14]. This second pulse is the Brillouin pulse, which functions thus: the generation of electrical charge through living tissue carries mechanical force. Many membranes have charged surfaces; there are dissociated ionic sites in proteins and DNA, and there are a host of chemical ions in tissue. All these are subject to these forces, so they, in turn, “radiate a portion of that energy as a propagating electromagnetic field” [15]. Therefore not only does the energy propagate more deeply than expected, but there is then an increased rate of collisions as they pass on their mechanical energy. This progression raises the total kinetic energy and, thus, the temperature of the medium as a whole. 

Albenese et al. point out that “deposition of thermal energy in a tissue can offset homeostatic mechanisms in the organism and cause physiological stress; at high temperature, deleterious phenomena such as protein denaturation can occur” [15].

In 1994, Albanese et al. raised the possibility of these electromagnetic transients being generated by modern transmitting devices, with regard to potential use for medical applications but also the creation of unintended adverse biological effects [16].

Instead of the normal exponential decay with distance, the Brillouin precursor pulse follows an algebraic amplitude decay of the peak amplitude. This changes the expected extent and depth of exposure; it does not increase the total amount of energy, but maintains some of it as kinetic energy, redistributes the frequency spectrum of the original signal and carries the energy deeper into the dispersive living tissue of the body than predicted by conventional thermal models [17]. Wet skin will result in the energy propagating deeper and the amplitude remaining higher [18]. 

Sommerfeld and Brillouin pulses only occur within media with di-electric properties, such as those of humans and animals. Calculations of their properties are affected by the nature of the tissue they enter—the temperature, the volumetric tissue density and tissue type. They will also vary according to propagation orientation, polarization and frequency. Further, ultra-wide band signals require consideration of all the frequencies involved [18].

Brillouin precursor pulses can be generated under certain circumstances. One of these occurs when transmission of sufficient energy is generated by very brief pulses using ultra-wideband RF (UWB) including mmW frequency radiation. This is relevant for 5G. Many mobile phones can already use UWB (e.g., iPhone series 11 to 14 and some Samsung models), and some Apple brand watches do so—bearing in mind these are worn against the skin. Download rates are still well below those promised, having so far achieved 432.7 megabits/s (Mbps) in South Korea, with a peak of 922.5 Mbps in Taiwan [19]. However, projected speeds will be 10 Gbps, with up to 20 Gbps technically possible. 

The likelihood of producing Brillouin precursors increases with extra transmission speed, increasing the pulse rate into Gbps (billion bits per second), and with a GHz bandwidth of more than 500 MHz. 

In 2019, Kurt Oughstun, a well-established researcher in this area, stated that it was unlikely that Brillouin precursors would be created by 5G technology, although, “a 10 Gbps (gigabits per second) data rate or higher would, however, be sufficient [to create Brillouin precursors], and that would be worrisome” (personal communication. Email discussion about precursor pulses with Kurt Oughstun, 2019, permission to use given 31 October 2021).

It seems likely that these conditions will be met within a few years, as in June 2022, the Global System Mobile Association (GSMA), the industry organisation representing the interests of mobile operators worldwide, published its policy position on the 5G spectrum [3]: 

“5G will be defined in a set of standardized specifications that are agreed on by international bodies–most notably the 3GPP [the 3rd Generation Partnership Project] and ultimately by the ITU [International Telecommunications Union]. The ITU defined criteria for IMT-2020—commonly regarded as 5G—and selected a set of compatible technologies which will support… **Enhanced mobile broadband**: Including peak download speeds of at least 20 Gbps…”p. 3 [original emphasis]

Beyond 5G and its peak download speeds in the gigabits range is 6G. According to a 2020 IEEE paper, the future introduction of the next-generation 6G communications will utilize much higher data speeds in the Terahertz band (0.3 THz to 10 THz), which the authors acknowledge “is the last unexplored band of the radio frequency (RF) spectrum” [20]. 

### 3.3. Research

At the time of submitting this paper, no in vitro or in vivo research had yet been published exploring heating or other adverse biological effects from exposures using 5G protocols. This is applied with or without the creation of Brillouin precursors with 5G phased array antennas (let alone with 6G communications). Since submission, two studies have been published and are added here during review. The first is a case report of two people who developed classic nervous and cardiovascular microwave syndrome symptoms after 5G transmitters were activated on the roof of their apartment, additional to 3G and 4G ones which had not caused problems. Exposure levels were between 39 and 278 times higher after 5G activation [21]. The second exposed rats to 250 μW/cm^2^ multi-frequency simulated 5G signals for 4 months, plus 1 month with no exposure. Serum levels of corticosterone and ACTH varied during exposure, but in both cases, were higher one month after exposure stopped than at any other point. The results indicated moderate stress [22]. We do not anticipate that Brillouin precursors were involved. 

Albenese et al. predicted in 1994 that the interaction of Brillouin precursor pulses with human tissue would cause the unwinding of large molecules, thermal damage and damage to cell membranes leading to blood–brain barrier leakage [15]. Oughstun considers “the most important effect [of Brillouin Precursors] is that radiation no longer decays exponentially in lossy materials such as water, foliage and biological tissue.… [G]roup velocity approximation… breaks down for pulses with short rise times… [In reality], the Brillouin precursors can become the dominant field,” becoming tens of times stronger at a depth that reflects back and forth inside the skull, leading to “several hot spots due to beam focusing” [23].

Most recently, research by Lawler et al. [24] describes and cites several studies encompassing exposures within and exceeding those of the ICNIRP guidelines and including the proposed 5G frequency range. Overall, they conclude there is “a strong power and dose dependence of MMW-induced effects at biologically relevant exposure levels”.

If conditions are insufficient to cause Brillouin precursor pulses, safety is not necessarily guaranteed. Specific absorption rates in the top layers are considerably higher at mmW frequencies. For instance, at 30 GHz and a skin depth of 0.782 mm and 5 mW/cm^2^, the SAR has been estimated as 65.5 W/kg [7]. For local exposures, 5 mW/cm^2^ is well within that permitted occupationally in the 2020 ICNIRP guidelines. The 1998 ICNIRP guidelines [25] allowed a maximum of 20 W/kg for occupational exposure of a limb to 10 MHz–10 GHz. Therefore, an acceptable exposure > 6 GHz measured on a surface (as required with the 2020 guidelines) may return an unacceptably high volume measurement (specific absorption rate) if that were assessed. However, it is not required in the ICNIRP 2020 guidelines or Australasian exposure standards. 

Extreme heat risk was predicted in December 2018 by Neufeld and Kuster [26]. They suggest that permanent skin damage from tissue heating may occur even after short exposures to 5G mmW pulse trains (where repetitive short, intense pulses can cause rapid, localized heating of skin). The authors stated that there is an urgent need for new thermal safety standards to address the kind of health risks possible with 5G technology. 

Neufeld and Kuster express the problem succinctly: 

“The FIFTH generation of wireless communication technology (5G) promises to facilitate transmission at data rates up to a factor of 100 times higher than 4G. For that purpose, higher frequencies (including mmW bands), broadband modulation schemes, and thus faster signals with steeper rise and fall times will be employed, potentially in combination with pulsed operation for time domain multiple access…The thresholds for frequencies above 10 MHz set in current exposure guidelines (ICNIRP 1998, IEEE 2005, 2010) are intended to limit tissue heating. However, short pulses can lead to important temperature oscillations, which may be further exacerbated at high frequencies (>10 GHz, fundamental to 5G), where the shallow penetration depth leads to intense surface heating and a steep, rapid rise in temperature…”.[26]

Neufeld, Samaras and Kuster followed this with a critically important paper. The authors are highly respected researchers working in the Foundation for Research on Information Technologies in Society (IT’IS), Europe’s leading electromagnetic energy research laboratory. After completing calculations, they published this warning. “In the case of pulsed narrow beams, the values for the time and spatial-averaged power density allowed by the proposed new guidelines could result in extreme temperature increases” [27]. They emphasise that “pulse-duration-independent limits on fluence (equivalent to limits on averaged power density in a given time interval) do not constrain the peak-to average ratio (or the peak) of the exposure” (p. 165). The authors clearly specify that an averaging area smaller than 4 cm^2^ is necessary to avoid overheating of tissue. Their calculations found possible increases of >4 °C up to a theoretical peak of “several 100 °C after 500 ms” (p. 167). 

The paper received criticism, in response to which they clarified that they did not anticipate imminent severe heating from current devices at the time of writing (2018), but that peak energy limits should be safe for future likely scenarios; they also clarified that with peak power densities, they were referring to “close near-field exposures… that can be well approximated by a Gaussian power distribution… rather than a very narrow, intentionally focused Gaussian beam” [28]. 

The final ICNIRP paper retains averaging over 4 cm^2^ and the need for averaging maximum absorbed energy density over 1 cm^2^, but only for exposures > 30 GHz and to the extent that it must not exceed twice the averaging value of 4 cm^2^ (p. 508). 

## 4. Discussion

By way of background, the ICNIRP is a self-governing private organization (NGO) that elects their members internally. Members have been criticized for having telecommunication industry ties and conflicts of interest with other work they have undertaken for the World Health Organisation [29]. Many assumptions on which the guidelines are based have recently been critiqued by the International Commission on the Biological Effects of Electromagnetic Fields and assessed as “failing to protect human and environmental health” [6]. The 1998 guidelines do not refer to averaging area at all, but rather averaging mass (i.e., SAR).

It seems quite possible that the heat threat has not been sufficiently addressed to ensure the guidelines are intrinsically safe. The Neufeld et al. paper, which was published in 2020 [27], was funded by unspecified government agencies and the telecommunication industry. Follow-up papers on this important topic, which could be expected, have not followed from these authors. In response to results reported herein, we propose that 5G local exposures should not only be assessed with absorbed power density (S_ab_ in the ICNIRP guidelines) and energy density for surface heat impact (both surface measurements), but should also require volume measurements over time. In other words, SA should also be applied for exposures > 6 GHz. 

This has been recommended elsewhere for obtaining an “effective and correct exposure for spread spectrum and ultrawideband signals” [30] such as those related to extremely high-frequency beamforming. 

Further research exploring both heat and biological impacts of 5G multi-frequencies and 5G with co-exposures should be an urgent priority. This necessity for more reliable in vitro and in vivo research into possible damaging effects of pulsed mmW used for 5G communications is emphasised by Foster and Vijayalaxmi [31] who analyse “31 studies related to genetic damage produced by exposure to RFR at frequencies above 6 GHz, including at millimeter-wave (mm-wave) frequencies. Collectively, the papers report many statistically significant effects related to genetic damage, many at exposure levels below current exposure limits”.

However, the authors then point out that the results in many of these studies may be non-replicable. They suggest that any conclusions from these findings are limited and insufficient for giving health advice or for setting exposure limits and, to address this problem, they call for improvements in study design, analysis and reporting in future bioeffects research. This, they consider, would provide more reliable information for health agencies and regulatory decision makers. 

While this is in line with epidemiological practice, those insisting on replicable results in this field often propose the need to use controlled (uniform, prescribed) exposures. The problem is that experience shows that controlled electromagnetic field exposures differ in nature to the high variability found in ‘real’ everyday exposures, and the consequent impacts on biological endpoints are typically lower. Despite varying exposure regimes, many common endpoints have been found in a large percentage of studies, so regarding this body of research as invalid seems risky.

However, we agree with the “take-home” message from the Foster and Vijayalaxmi paper that we still do not have adequate research on 5G mmW to be able to assure the public that the many thousands of 5G antennas, in many instances placed very near homes and workplaces, are without a possible health risk because the necessary research has not yet been conducted. 

While the thermal impact of mmW is the most noticeable, it is possibly not the most harmful. Effects other than thermal effects have been shown to occur at levels much lower than those required to raise the temperature of living tissues significantly. Chronic or recurring exposures may be worse, and strongest non-thermal responses can occur days or weeks after exposure ends [22] although the majority of studies fail to evaluate this.

There is ongoing concern regarding adverse bio-effects, and there is no reason to believe these will be fewer with 5G added to the mix. Impacts on skin have been found in previous research. A recent critique of the 99 mmW studies on skin and skin cells that have been published reports that none examined possible delayed or long-term exposure effects, and none considered co-exposures. The author concludes that the research status is insufficient to “develop science-based human health policies” [32]. The only human 5G study published reports skin problems after nearby 5G installation [21]. Many other bio-effects have been found repeatedly; the range and frequency of these is reported by Leach et al. [33], who report that biochemical changes and signs of oxidative stress are those most commonly observed. The added frequency range that comes with broadband 5G increases the likelihood of there being frequency-specific responses. It is possible some may be advantageous, but this overlooks the fact that some may trigger adverse effects or exacerbate existing conditions. Again, research is needed.

A report commissioned by the European Parliament’s Global Committee on Industry, Research and Energy emphasizes the complexity of 5G and concludes that its “unpredictable propagation patterns that could result in unacceptable levels of human exposure to electromagnetic radiation” (p. 6) [34]. This is challenging for those in standards organisations who need to prepare and incorporate exposure specifications suitable for 5G. This also affects standard-setting of how to undertake laboratory and environmental measurements. Hence, the report’s foremost recommendation is to increase long-term research and development to allow better understanding, measurement and control of exposures at mmW frequencies (pp. 27–28). Early approaches have been published on methods to evaluate these highly complex exposures in the environment [35,36] and in the laboratory [37]. 

## 5. Conclusions

Surface radiofrequency exposure assessments including mmW radiation are insufficient to ensure safety; there are several reasons assessment of SA_ab_ is also needed.

A real danger of the ‘expert’ assurances of a lack of risk is that they discourage the necessary research to evaluate risk properly. They may also discourage review of apparently outmoded/questionable approaches being taken in RF exposure standards.

Once the 5G mmW band is internationally operational, a significant proportion of the world’s population will be exposed to new hazards. The intensity and complexity of near-field exposure, such as when carrying a phone in a pocket or using it next to the head, will be different for 5G, and this is the first time mmW have been used for public telecommunications and the first time beamforming has been deliberately introduced for near-field use. Without research on the impact of near-field 5G, this global step is an experiment at the population level. Bearing this in mind, there is a vital and urgent need for targeted research and for a re-evaluation of the scientific relevance of the current RF human exposure standards’ basic approach and assumptions.

## Figures and Tables

**Table 1 ijerph-20-05267-t001:** Average depth microvasculature and free nerve endings.

Depth of Upper Cutaneous Blood Vessel from Skin Surface (mm)	Depth of Upper Cutaneous Blood Vessel from Skin Surface Sensitive Skin (mm)	Approx. Depth of Preauricular Free Nerve Endings from Healthy Skin Surface (mm)
1.15–1.45	0.19–0.23	0.025 ^a^
(Cicci et al., 2016, pp. 269–280) [10]	(Jiang et al., 2020, pp. 431, 433) [11]	(Lintzeri et al., 2022, p. 1194 table 1) [12]

^a^ Lintzeri et al. (2020) systematic review of six studies found the mean epidermal thickness in front of the ear (preauricular) was 0.049 mm (49.2 µm) so we take the approximate depth of the free nerve endings in the third of five epidermal layers here to be 0.025 mm. Jiang et al. (2020) showed that those with sensitive skin tend to have shallower facial microvasculature ranging from 0.189 mm to 0.231 mm.

## Data Availability

No new data were created or analysed in this study. Data sharing is not applicable to this article.

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
