# Peer review of "ICNIRP Guidelines’ Exposure Assessment Method for 5G Millimetre Wave Radiation May Trigger Adverse Effects"

_ijerph, 2023, doi:10.3390/ijerph20075267_

Round 1

Reviewer 1 Report (New Reviewer)

The manuscript entitled ”Are there risks of heat-damage or other introduced bio-effects from 5G ICNIRP permitted exposures?” was reviewed.

In this study, the authors aimed to examine whether the new approach of ICNIRP is satisfactory to prevent heat damage and other possible bio-effects once millimeter wave 5G is included. 5G is an important issue in the radiation field and there are important messages in this manuscript, but these messages have already been mentioned in other articles (for example: “International Commission on the Biological Effects of Electromagnetic Fields (ICBE-EMF). Scientific evidence invalidates health assumptions underlying the FCC and ICNIRP exposure limit determinations for radiofrequency radiation: implications for 5G. Environ Health 21, 92 (2022). https://doi.org/10.1186/s12940-022-00900-9”.

Recommendations:

The innovations of the article need to be mentioned.

The hypothesis and study question are not clear.

The method and type of study are not stated.

Also, the manuscript is not well-structured

Other recommendations:

1.      Title: the title is not informative and is suitable for experimental studies. While this manuscript is not experimental.

2.      Introduction:

o   the aim of the study is not clear. I think the hypothesis of this manuscript could be published as a review article or letter to the editor or brief report but their structure is not followed.

o   What is the innovation of this article? and explain the differences and similarities with other articles.

3.      Line 95: “This has been reinforced by……” is not clear.

4.      Discussion: The need for discussion in this manuscript is not clear to me. No data were obtained to discuss and a short conclusion seems to be sufficient.

5.      Because of the lack of in vitro and in vivo study about the 5G, its possible health hazards is not clear and we cannot doubt the existing standards with certainty. It should also be noted that existing standards need to be updated, as with ionizing radiation standards, over time.

6.      The conclusion is not written correctly. Most of this section is the discussion.

7.      Author Contributions: line 316 is repeated.

8.      Institutional Review Board Statement –Data Availability Statement  Also Informed Consent Statement:: is copy-pasted of author instruction.

Author Response

Please see our response attached. 

Reviewer 2 Report (New Reviewer)

The manuscript consists of total 8 pages, including the list of total 28 literature references. The article reviews the problem of possible negative tissue heating-related health effects of exposure to beam-type ultra-short wave irradiation emitted from the modern 5G wireless communication infrastructure that are still within the limits allowed by the newest international guidelines. As such, the article is current and likely to rise the interest in the Readers.

The Abstract mirrors the key contents of the main text of the article adequately.

The Introduction provides enough background information concerning the reviewed problem.

The results are presented in a logical manner. In my opinion, the phenomena Sommerfield and Brillouin need to be explained in a more detailed manner and separately one after another, preferably in separate text sections, contrasted against each other, so the Readers can understand their nature and the difference between them more easily. In my opinion, the objective data shall be presented first and then followed by a separate section containing the concerns of the Authors, including the views about the objectivity and impartiality of the expert body setting standards shall be collected and discussed in a single, separate section in the end of the text (discussion) - instead of being scattered in several places through the article (in the former part the style shall be objective and reporting while in the latter the style may be more argumentative and narrative - while now the Authors mix these a lot freely, which is not advisable); this would preserve the sound scientific approach that divides the presentation of facts from their interpretation by the Authors.

The Discussion section shall contain its current contents plus the argumentative and subjective fragments from other sections of the text, including also a big part of the current long and narrative Conclusions. There shall be a more pronounced remark that the thermal effect is the most noticeable but possibly not the most harmful effect of radio-frequency exposure, while other than thermal biological effects may occur after exposures much lower than those required to rise the temperature of living tissues significantly, especially if the exposures are of chronic or recurring nature.

The Conclusions need to be rewritten as now this section contains many elements of argumentation that belong in the Discussion section. On the other hand, the article lacks the typical - and expected - concise and informative at the same time conclusions.

The topic begs for some graphic figures illustrating the presented problems, which highly unfortunately are missing in the current version of the article.

The literature references are numerous and recent enough. The Authors may want to enrich the scope of the introductory remarks by tackling the following aspects:
- chronic exposure risks, e.c. https://doi.org/10.3390/ani11092721
- exposure from sources currently already in operation, e.c. https://doi.org/10.3390/app11083592
- exposure measuring attempts methodology, e.c. https://doi.org/10.3390/app11041751 , https://doi.org/10.3390/app10175971 , https://doi.org/10.3390/app10155280 , https://doi.org/10.3390/environments7030022 , https://doi.org/10.3390/electronics9020223
- exposure biological effect, e.c. https://doi.org/10.3390/ijms22031342 , https://doi.org/10.3390/ijerph15102320

Author Response

Our response is attached. 

Reviewer 3 Report (New Reviewer)

This manuscript can be recommended for publication after taking into account the following comments:

1. Line 215 needs reference support.  Note that Samaras is not  first author of any cited papers in the reference list.

2. Line 102-106. A very incomprehensible paragraph. Give more information with relevant links.

Round 2

Reviewer 1 Report (New Reviewer)

The article entitled” ICNIRP Guidelines' exposure assessment method for 5G millimeter wave radiation may trigger adverse effects” was re-reviewed. It has improved a lot in terms of quality, but the following things need attention:

1- The purpose of the study should be mentioned in the last paragraph of the introduction.

2- Please see the "Types of Publications" section in the Instructions for Authors. I suggest these headings: Abstract, Keywords, Introduction, ICNIRP Guidelines, Brillouin precursors, Research, Discussion, and Conclusions.

3- In table 1, the third and fourth columns have repeated numbers and you can combine the cells of this column and write a number.

4- Please correct this paragraph and remove the duplicated part:

"Author Contributions: Author contributions: Each author contributed equally to the paper and has …..”

Author Response

Thank you for these further suggestions. Responses are provided after each point, as before.

The article entitled” ICNIRP Guidelines' exposure assessment method for 5G millimeter wave radiation may trigger adverse effects” was re-reviewed. It has improved a lot in terms of quality, but the following things need attention:

  • The purpose of the study should be mentioned in the last paragraph of the introduction.

Response: We have added the following sentence at the end of the introduction: “This paper explores the assumption that it is not necessary to assess 5G exposures > 6 GHz using Specific Absorption (SA).”

  • Please see the "Types of Publications" section in the Instructions for Authors. I suggest these headings: Abstract, Keywords, Introduction, ICNIRP Guidelines, Brillouin precursors, Research, Discussion, and Conclusions.

Response: We agree. These are the headings we had in the original submitted paper. The “ICNIRP Guidelines, Brillouin precursors, Research” all fit within the Results section in line with another reviewer’s suggestion. Your first review requested a Method section which we also include.

  • In table 1, the third and fourth columns have repeated numbers and you can combine the cells of this column and write a number.

Response: Done

4- Please correct this paragraph and remove the duplicated part: 

"Author Contributions: Author contributions: Each author contributed equally to the paper and has …..”

Response: Done

We have also added “trigger adverse effects” in lines 338-339 to tie it to the new title: “It is possible some may be advantageous but overlooks that some may trigger adverse effects or exacerbate existing conditions. Again, research is needed.”